# Neonatal Brain Abscess with *Serratia marcescens* after Intrauterine Infection: A Case Report

**DOI:** 10.3390/antibiotics12040722

**Published:** 2023-04-07

**Authors:** Mihaela Bizubac, Francisca Balaci-Miroiu, Cristina Filip, Corina Maria Vasile, Carmen Herișeanu, Veronica Marcu, Sergiu Stoica, Catalin Cîrstoveanu

**Affiliations:** 1Department of Neonatal Intensive Care, “Carol Davila” University of Medicine and Pharmacy, 020021 Bucharest, Romania; 2Neonatal Intensive Care Unit, “M.S. Curie” Children’s Clinical Hospital, 041451 Bucharest, Romania; 3Department of Pediatrics, “Carol Davila” University of Medicine and Pharmacy, 020021 Bucharest, Romania; 4Department of Pediatric Cardiology, “M.S. Curie” Children’s Clinical Hospital, 041451 Bucharest, Romania; 5Department of Pediatric and Adult Congenital Cardiology, University Hospital of Bordeaux, 33600 Bordeaux, France; 6Ph.D. School Department, “Carol Davila” University of Medicine and Pharmacy, 020021 Bucharest, Romania; 7Department of Radiology, ”M.S. Curie” Children’s Clinical Hospital, 041451 Bucharest, Romania; 8Department of Neurosurgery, Monza Hospital, 021967 Bucharest, Romania

**Keywords:** brain abscess, neonate, *Serratia marcescens*, sepsis, assisted human reproduction techniques, high-risk pregnancy, antibiotic treatment, multidrug-resistant bacteria

## Abstract

Brain abscesses are a possible complication of bacterial sepsis or central nervous system infection but are uncommon in the neonatal period. Gram-negative organisms often cause them, but Serratia marcescens is an unusual cause of sepsis and meningitis in this age group. This pathogen is opportunistic and frequently responsible for nosocomial infections. Despite the existing antibiotics and modern radiological tools, mortality and morbidity remain significant in this group of patients. We report an unusual unilocular brain abscess in a preterm neonate caused by *Serratia marcescens*. The infection had an intrauterine onset. The pregnancy was achieved through assisted human reproduction techniques. It was a high-risk pregnancy, with pregnancy-induced hypertension, imminent abortion, and required prolonged hospitalization of the pregnant woman with multiple vaginal examinations. The infant was treated with multiple antibiotic cures and percutaneous drainage of the brain abscess associated with local antibiotic treatment. Despite treatment, evolution was unfavorable, complicated by fungal sepsis (*Candida parapsilosis*) and multiple organ dysfunction syndrome.

## 1. Introduction

The healthcare environment is conducive to transmitting pathogens which are ten times more resistant to treatment than average, including *Serratia marcescens* [1].

Although originally described in 1913, its prevalence was underestimated in human disease for a long time until the first reported outbreak of nosocomial Serratia marcescens infection was reported in 1951. Since then, infections due to this organism have been increasingly reported [2,3].

*S. marcescens* is the main pathogen responsible for central nervous system infections, respiratory tract infections, bacteremia, endocarditis, peritonitis, arthritis, osteomyelitis, keratitis, urinary tract infections, and skin infections [4].

*Serratia* spp. has been reported as the third most frequent pathogen in neonatal facility outbreaks [5]. Environmental sources have been described (e.g., contaminated medical devices, parenteral nutrition, contaminated milk, intravenous and topical solutions, liquid soap, respiratory care equipment, and air conditioning) [6,7,8,9]. Serratia can cause invasive systemic infections in newborns due to the immaturity of the immune system [10,11,12]. 

Bacteremia and sepsis can often lead to central nervous system infections in neonates [13]. Brain abscesses are a possibly life-threatening complication of bacterial sepsis or central nervous system infection. Still, they are uncommon in the neonatal period [14], occurring in about 1.3–4% of neonates with meningitis [15], and are usually associated with high mortality and morbidity. 

A brain abscess is a focal, intracerebral infection that develops into a collection of purulent exudates surrounded by a vascularized capsule [16]. *Streptococcus* spp. is now the most identified pathogen responsible for brain abscesses. At the same time, *S. marcescens* has been described in only a few reports; these were cases of late neonatal sepsis [17,18]. 

Concerning intrauterine abscesses, Volpe outlines that the bacteria cause meningitis and vasculitis and later complicate it, resulting in brain abscess formation. In this case, there is usually a history of amniotic fluid leakage for a few days before delivery. This is followed by neonatal sepsis and hematogenous dissemination of the brain parenchyma, which results in multiple abscesses with no initial meningeal involvement [19]. Moreover, the colonization of the brain parenchyma during sepsis could be explained by the physiological right-to-left shunting of the neonatal circulation [20]. 

We present the case of a single brain abscess in a preterm neonate as a complication of *Serratia marcescens* meningitis after intrauterine infection. 

## 2. Case Presentation

We present the case of a male neonate, delivered by cesarean section at 35 weeks’ gestation, with a weight of 2300 g and an Apgar score of 9, due to a retroplacental hematoma in a high-risk pregnancy (in vitro fertilization, gestational hypertension). The pregnant woman was hospitalized for almost the entire pregnancy due to pregnancy-induced hypertension and impending abortion. She required multiple medical examinations and treatments. In the first hours of life, the newborn presented with generalized edema and tonic–clonic seizures on the second day.

The patient was transferred to the Neonatal Intensive Care Unit at “Marie S. Curie” Children’s Hospital, Bucharest, at three days of life. His overall condition was severe; he was febrile, intubated, and mechanically ventilated, with a weight on admission of 2120 g. His vital signs were SpO2 = 96%, HR = 150 bpm, BP = 60/35/40 mmHg, and three seconds capillary refill time, with generalized edema, convulsive status epilepticus, hepatosplenomegaly, low axial tone, and normotensive anterior fontanelle.

A complete blood count showed thrombocytopenia (16.000/mm^3^); positive C-reactive protein 153.62 mg/dL (*n* < 5 mg/dL) and procalcitonin > 10 ng/mL; and hyponatremia (129 mmol/L) and hypocalcemia (0.97 mmol/L). The cardiac ultrasound upon admission was standard, with a closed patent ductus arteriosus. The chest X-ray did not reveal any abnormalities.

The cranial ultrasound upon admission showed a right-sided frontoparietal hyperechogenic mass of about 19/20/30 mm (Figure 1 and Figure 2). The cranial CT confirmed a right frontoparietal cerebral abscess (20/23/20 mm) compressing the frontal horn at the right lateral ventricle, as well as moderate ventricular system dilation (Figure 3).

Cerebrospinal fluid analysis, collected through lumbar punction, revealed normotensive, xanthochrome, and cloudy cerebral spinal fluid with a positive Pandy’s reaction, a total cell count of 157,000 /mm^3^ (95% polymorphonuclear cells, 5% lymphocytes), glycorrhachia 10 mg/dL, and CSF total protein 801.9 mg/dL. The CSF culture was positive for *Serratia marcescens*. The diffusimetric antibiogram was performed according to the EUCAST (European Committee on Antimicrobial Susceptibility Testing) guide from 2010 using a Vitek2 Compact device (BioMerieux, Lyon, France). The results are shown in Table 1. Hemocultures and tracheal aspirate cultures were also positive for the same multidrug-resistant Serratia marcescens (Table 1). 

After admission, a central venous catheter was placed through the right femoral vein. Immediate therapy with intravenous antibiotics was initiated (Meropenem 40 mg/kg/dose every 12 h and Vancomycin 15 mg/kg/dose every 12 h). When the results of the cultures from the CSF, blood culture, and tracheal aspirate were positive for *Serratia marcescens*, Vancomycin was changed to Netilmicin, 4 mg/kg/day, for a period of 10 days, and then Ciprofloxacin 20 mg/kg/day for another four days. Antifungal therapy was initiated with Fluconazole at a therapeutic dose for eight days, followed by prophylactic therapy. The convulsive treatment was performed with Phenytoin, 15 mg/kg/dose with a single dose, and then 6–8 mg/kg/day in two divided doses. Intermittently, he also received Diazepam. Dexamethasone was added to the treatment for its anti-inflammatory effect (4 days). 

The effectiveness of the treatment was monitored by CSF analysis. Initially, CSF was obtained by lumbar puncture (the first two determinations). Later, a ventricular puncture was performed. Ventricular endoscopy showed an ependymal fistulized brain abscess. The abscess was aspirated, and external ventricular drainage was mounted on the 16th day after admission. *Serratia marcescens* was also isolated from the abscess culture. We started intraventricular treatment with Chloramphenicol (10 mg/kg/dose daily). 

The evolution was slowly favorable. The patient showed no signs of fever from the fourth day of treatment. Mechanical ventilation was necessary for 16 days because spontaneous breathing is impossible in severe neurological illnesses. The convulsive manifestations ceased, and Phenytoin was progressively discontinued from the treatment during the following 24 days. A cystic cavity was formed in place of the abscess that communicated with the right lateral ventricle (Figure 4).

Parenteral nutrition was initiated on admission through a central venous catheter, and from day ten after admission, the patient received enteral nutrition with breast milk through a feeding tube.

An improvement in the laboratory parameters was also found, with a progressive decrease in CSF total protein and normalization of the complete blood count and inflammatory markers (Figure 5 and Figure 6). However, one month after admission, the patient presented with fever, anemia, thrombocytopenia, and generalized tonic–clonic seizures.

The antibiotic treatment scheme was changed (intravenous Aztreonam 30 mg/kg/dose every 8 h, Amikacin 10 mg/kg/dose every 8 h). The intraventricular treatment was replaced with aminoglycosides, Gentamicin 2 mg/kg/day for four days, and Amikacin 5 mg/kg/dose daily. Anticonvulsant therapy was reintroduced (Phenytoin, Diazepam, Thiopental). The central venous catheter on the right femoral vein was also removed and replaced with a central venous catheter through the right internal jugular vein. The external ventricular drainage was eventually replaced. The patient required multiple blood transfusions, platelet mass, and fresh frozen plasma.

His clinical condition progressively deteriorated, presenting with generalized edema and oliguria. He received diuretic treatment, initially intermittent and later in continuous infusion. On the 41st day after admission, the patient was initiated on peritoneal dialysis, which remained functional for nine days. However, he developed abdominal compartment syndrome and required the removal of the peritoneal catheter. During the peritoneal dialysis, he presented with multiple episodes of hyperglycemia and was started on intravenous continuous insulin infusion for four days. 

The central venous catheter tip culture and blood cultures isolated *Candida parapsilosis*, for which he received Caspofungin for 14 days and Amphotericin B for three days. He continued to receive intraventricular treatment with Amikacin for 25 days, combined with intravenous administration of Piperacillin tazobactam (5 days), Aztreonam (18 days), Imipenem-Cilastatin (8 days), Cefepime (10 days) and Linezolid (5 days). 

The patient’s general state was progressively degrading. He presented with abdominal compartment syndrome, which required changing the ventilation mode to high-frequency oscillatory ventilation. The patient became anuric, showing massive edema, bradycardia, and hypotension, for which he needed a continuous infusion of inotropic agents at high doses (Dobutamine up to 15 µg/kg/min, Dopamine up to 15 µg/kg/min, Adrenaline up to 2 µg/kg/min). On the 54th day after admission, the patient was declared dead.

The anatomopathological result obtained through autopsy revealed old ventricular hemorrhage, ventriculitis, and significant hydrocephalus.

## 3. Discussion

Serratia marcescens is ubiquitous and not a standard component of human fecal flora. Most infections are acquired exogenously [21]. A recent analysis of the causes of neonatal sepsis in developing countries cited the *Serratia* species as the fifth most common cause (0.5% for early neonatal sepsis and 0.3% for late neonatal sepsis) [22]. 

There have been outbreaks of *Serratia marcescens* in neonatal intensive care units described in the literature [2,5,6,7,8,9,10,11,23]. Still, they often did not progress to the central nervous system or other complications. Usually, *S. marcescens* and cerebral infections are linked to cerebral instrumentations, such as neurosurgical procedures or lumbar punctures. Sometimes, a congenital central nervous system malformation (e.g., myelomeningocele) can predispose one to infection [17]. Few cases of brain abscesses after an initial diagnosis of *S. marcescens* septicemia and meningitis have been published in the literature [4,18,21,24].

We report the case of a premature newborn preterm infant (35 weeks’ gestation, birth weight 2300 g) who developed signs of meningitis on the second day of life (fever, severe general condition, generalized tonic–clonic seizures). This clinical pattern is suggestive of a maternal–fetal infection [25].

Systemic Serratia infection could be the result of Serratia chorioamnionitis infection. No data are available regarding the mother’s medical conditions at delivery. The baby’s prognosis was severe, with high mortality in such situations. In 2021, Ottolini et al. published the case of a premature infant who survived chorioamnionitis with Serratia. Until then, it was the only such case in the literature [26]. Chorioamnionitis with Serratia can cause premature rupture of the membranes and the birth of a premature fetus, most frequently with systemic infection or pregnancy loss [27]. The source of infection was presumed to be urinary infection, venous line sepsis, or direct inoculation post-chorionic villus sampling. In other cases, the source was a premature rupture of the membranes and ascending infection from vaginal colonization. Therefore, it is essential to avoid multiple vaginal examinations in order to minimize the risk of intrauterine infection [27]. Assisted human reproductive technology is a possible cause of premature rupture of membranes [28]. Previous studies have reported that assisted reproductive technology is associated with an increased incidence of several pathologies, such as hypertension during pregnancy, gestational diabetes mellitus, intrahepatic cholestasis of pregnancy, postpartum hemorrhage, preterm birth, and low birth weight [29].

Imaging methods were crucial in establishing the diagnosis of a brain abscess. Cranial ultrasound is the first non-specific diagnostic technique for diagnosing abscesses, as it can be used at the bedside without sedation and can be repeated if necessary. However, its sensitivity is not high in parenchymal lesions that are not hemorrhagic [30]. Computed tomography is commonly used to diagnose brain abscesses in adults because it has high specificity after contrast enhancement. Furthermore, CT scanning is more useful in older children than in newborns because neonates have a higher water capacity in the brain, which reduces the contrast between normal and affected tissues [31]. Weber et al. describe MRI as having the highest sensitivity and specificity for diagnosing brain infections and abscesses and detecting very early stages of cerebritis [32]. However, few reports prove its usefulness in diagnosing brain abscesses in neonates. In addition, it has significant disadvantages, such as the need to move the patient for a longer period and a longer duration of sedation [31]. The MRI was not used for our case patient due to its lack of availability in our clinic at that time. After establishing the brain abscess diagnosis, anticonvulsant and antibiotic treatment started immediately. 

The antimicrobial agents used to treat the central nervous system need to penetrate the blood–brain barrier and achieve the necessary concentrations in the CNS to eradicate the infecting pathogen [14,33]. Different strains of *S. marcescens* responsible for epidemic events have proven to be multi-resistant to antibiotic drugs. This species exhibits intrinsic resistance to βlactams and tetracyclines, but is susceptible to other antimicrobials, such as quinolones and aminoglycosides [34,35]. Fourth-generation cephalosporin, carbapenems, and piperacillin-tazobactam are currently being implemented as treatment options [36]. One study suggested that a prolonged infusion of meropenem for three hours, divided into doses, would yield better CNS penetration with promising results [37]. For our patient, we used Meropenem, 40 mg/kg/dose, every 12 h for the first seven days of life, then every 8 h, in a 1-hour infusion. Usually, the antibiotics course would last 6–8 weeks [38]. Lately, carbapenemase-producing strains of *S. marcescens* have been found more frequently. This represents a severe risk because this organism is intrinsically resistant to colistin, which is considered the last drug option for carbapenem-resistant infections [39]. 

The management of brain abscesses is influenced by several factors, such as the patient’s neurological status, the abscess’ location, and the number and sizes of the abscesses if multiple are present [40]. Neurosurgical consultation must always be obtained, including abscess drainage when possible. This procedure is recommended if the abscesses are more extensive than 2 cm in diameter [18]. Recent opinions have emerged that ultrasonic aspiration is more effective in completely removing purulent and dense material [41].

In our study, a second follow-up CT scan was performed seven days after admission, revealing an increase in the right frontal hypodense area with a midline mass effect. Based on this evidence, it was decided that the brain abscess should be percutaneously drained. An external ventricular drain was inserted, and the patient received daily treatment with chloramphenicol. Several studies have confirmed that the topical use of antibiotics in the abscess cavity significantly increased the local concentration of the drug. This drug infiltrates the cavity wall and, therefore, targets the abscess at this level [42].

Steroid use to treat brain abscesses in neonates remains controversial. Several studies have suggested that corticosteroids may delay encapsulation and limit leukocyte migration for pus formation [43].

Initially, the evolution was favorable, and the patient started to receive enteral nutrition. Fever and convulsive manifestations disappeared, allowing for the progressive withdrawal of Phenytoin. We monitored the CSF cultures and the protein values in the CSF and glycorrhachia. Although there was a permanent decrease in the protein values in the CSF, the prognosis was poor because there were extremely high protein values in the CSF, up to 800 mg/dL (Figure 6). Tan J et al. [44] stated that a high CSF protein concentration may prognosticate poor outcomes in neonates with bacterial meningitis. The best cut-off for predicting a poor outcome was 188 mg/dL in CSF protein concentration (sensitivity 70.8%, specificity 86.2%) [41]. Glucose CSF is another important parameter. In general, a CSF glucose concentration < 30 mg/dL (1.7 mmol/L) in a term infant or <20 mg/dL (1.1 mmol/L) in a preterm infant is consistent with bacterial meningitis. The ratio of CSF to serum glucose is not useful in acutely ill neonates (their serum glucose may be increased secondary to stress or administration of intravenous glucose before the time of evaluation) [45]. Generally, low CSF glucose concentration persistence is an important prognostic factor in poor outcomes [44].

Systemic Candida infection is common in neonatal intensive care units, although the patient from this report received prophylactic antifungal treatment. *Candida parapsilosis* infections are responsible for one-third of neonatal Candida infections and are associated with high morbidity and mortality [45]. Neonatal risk factors for invasive *Candida parapsilosis* infections are low birth weight, prematurity, parenteral nutrition, central venous lines, catheters or other tubes, major surgery, and the use of broad-spectrum antibiotics, steroids, or H2 blockers [46]. The management, in this case, was that which was recommended, although the catheter and treatment with Caspofungin and Amphotericin B were changed [47]. Unfortunately, the multi-organ damage led to the death of the patient.

Candida infection precipitated the death of this patient, who had a poor prognosis in any case. A brain abscess identified in the first week of life is definitely due to an intrauterine infection. It is impossible to specify with certainty the moment of acquiring the intrauterine infection or the moment of the formation of the abscess. It could have been during the invasive maneuvers performed on the pregnant woman during her prolonged hospitalization, but infection during the assisted human reproduction procedure cannot be excluded. Data from the literature indicate that the source of most Serratia marcescens outbreaks remains unknown. This fact can be explained based on the ubiquity and resistance of this species; it can survive on inanimate surfaces (including medical devices) and even in antiseptic solutions for long periods. In addition, they colonize the respiratory and gastrointestinal tracts of both symptomatic and asymptomatic hosts [48].

## 4. Conclusions

Neonatal brain abscesses are challenging to treat, and have a poor prognosis and significant neurological sequelae if the patient survives. A brain abscess due to an intrauterine infection is unusual, and *Serratia marcescens* is an uncommon pathogen for vertical disease transmission. 

## 5. Teaching Point

Precaution is needed in any procedure on a pregnant woman. A nosocomial infection may have late, often fatal, consequences for the fetus. 

Early diagnosis of infected or contaminated patients and prompt and effective implementation of infection control measures are critical factors in limiting the spread of *S. mercescens.*

## Figures and Tables

**Figure 1 antibiotics-12-00722-f001:**
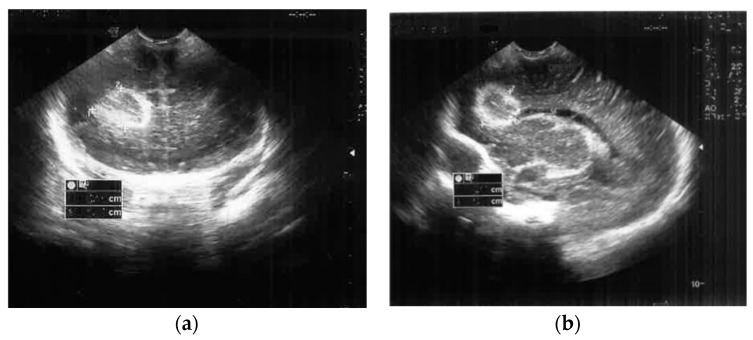
Brain ultrasound on the second day of life ((**a**) Frontal coronal plan, (**b**) right parasagittal plan) showed a nodular frontal lobe lesion with a hyperechoic demarcation line.

**Figure 2 antibiotics-12-00722-f002:**
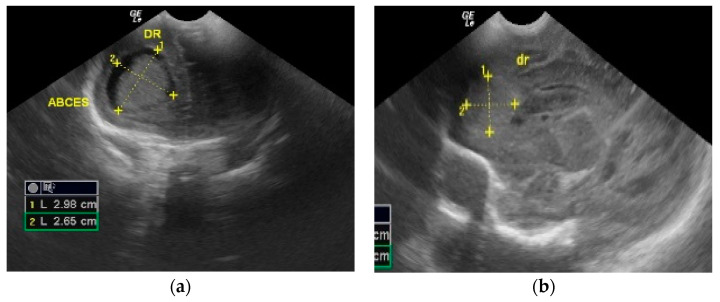
Brain ultrasound on the 7th day of life performed in 2 sections: (**a**) Frontal coronal plan, (**b**) right parasagittal plan) that showed a nodular frontal lobe lesion without a proper wall identifiable by ultrasound and with heterogeneous, suggestive necrotic content.

**Figure 3 antibiotics-12-00722-f003:**
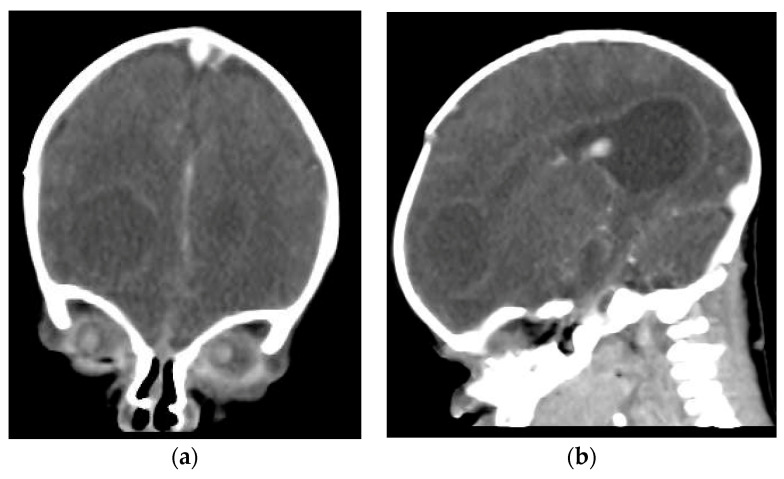
Cerebral CT on the 7th day of life ((**a**) Coronal plan, (**b**) sagittal plan) confirmed a necrotic right frontal brain lesion, associated with peripheral linear contrast enhancement (brain abscess). In addition, the CT scan identified the widening of the right LV (**b**) and the ventricular meningeal contrast enhancement (meningeal infection).

**Figure 4 antibiotics-12-00722-f004:**
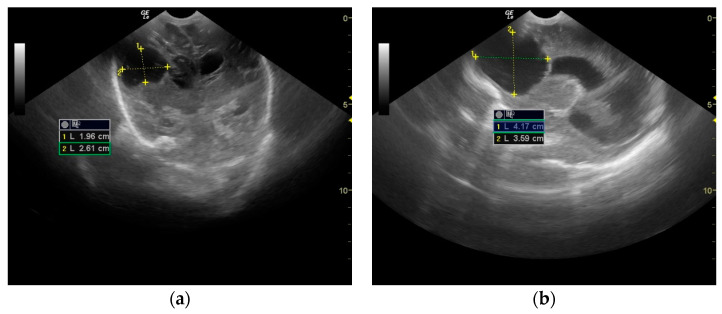
Brain ultrasound on the 30th day of life: cystic cavity ((**a**) Frontal coronal plan, (**b**) right parasagittal plan).

**Figure 5 antibiotics-12-00722-f005:**
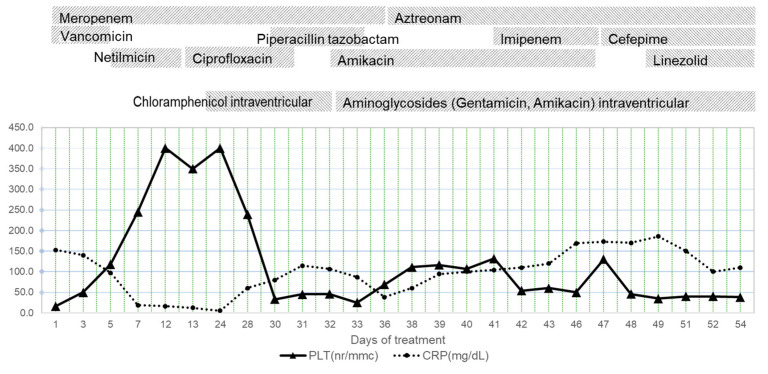
Evolution of the blood tests during hospitalization. CRP—C-reactive protein, PLT—number of platelets.

**Figure 6 antibiotics-12-00722-f006:**
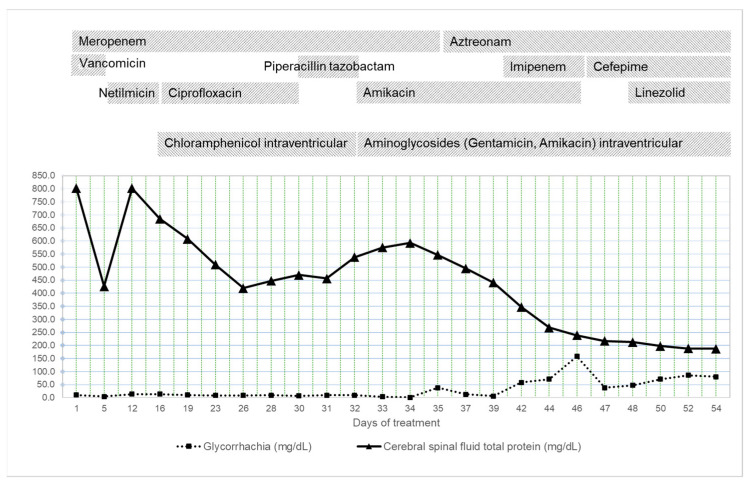
Evolution of glycorrhachia and total protein in cerebral spinal fluid during hospitalization.

**Table 1 antibiotics-12-00722-t001:** Antibiotic susceptibility of *Serratia marcescens* in our case.

Antimicrobial Agent	Sensitivity
Ciprofloxacin	S
Trimethoprim-sulfamethoxazole	S
Imipenem	S
Meropenem	S
Amikacin	I
Piperacillin-tazobactam	I
Amoxicillin	R
Ampicillin	R
Cefoperazone +/− Sulbactam	R
Ceftrixaone	R
Cefuroxime	R
Gentamicin	R

R = resistant; S = sensitive; I = intermediate.

## Data Availability

Not applicable.

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
