# Peer review of "Neonatal Brain Abscess with Serratia marcescens after Intrauterine Infection: A Case Report"

_antibiotics, 2023, doi:10.3390/antibiotics12040722_

Round 1

Reviewer 1 Report

Thank you for the opportunity to review this interesting manuscript about Serratia marcescens brain abscess. This study highlights the importance of this pathogen as a cause as an etiologic agent of central nervous system infections. Please consider the following comments:

ABSTRACT: please write Candida parapsilosis in italic letters

CASE PRESENTATION: There are some points where the reader has difficulty deducing the chronology in which antibiotics were used. Integrating the antibiotic schedules into a table could be very useful.

Line 85, page 3: The term "a convulsive state" is confusing. I suggest modifying the term to convulsive status epilepticus or convulsive seizures

  • Line 113, page 4: I suggest modifying the term "multi-resistant to antibiotic drugs" to Multidrug-Resistant Serratia marcescens.

    Line 126, page 4: the phrase "seizure manifestations were interrupted" is confusing, the authors are possibly referring to anticonvulsant drugs.

    Line 173, page 6: I found the phrase “the general stante was progressively degrading” confusing.  I suggest modifying to “his clinical condition progressively deteriorated”.

DISCUSSION: This section is in need of some English language editing. There are some sections where the reader has difficulty undertanding

The discussion repeats phrases written in the introduction. I recommend reducing redundant phrases.

  • Line 185, Page 6: In relation to the ecological niche of the bacteria "everywhere" is not the correct scientific term. "Ubiquitous" is the correct term.

  • Line 254, Page 8: The sentence begins with a lowercase n, please correct this sentence.

  •  
  •  
  •  

Author Response

Dear reviewer,

We appreciate your comments. We have performed the modification according to your suggestions.

ABSTRACT: please write Candida parapsilosis in italic letters

We modified it accordingly. 

CASE PRESENTATION: There are some points where the reader has difficulty deducing the chronology in which antibiotics were used. Integrating the antibiotic schedules into a table could be very useful.

We improved the presentation of our case report following your suggestions. Data regarding the antibiotics and dosage have been added.

Line 85, page 3: The term "a convulsive state" is confusing. I suggest modifying the term to convulsive status epilepticus or convulsive seizures

We modified it accordingly.

Line 113, page 4: I suggest modifying the term "multi-resistant to antibiotic drugs" to Multidrug-Resistant Serratia marcescens.

We modified it accordingly.

Line 126, page 4: the phrase "seizure manifestations were interrupted" is confusing, the authors are possibly referring to anticonvulsant drugs.

We rephrased. “The convulsive manifestations ceased, and Phenytoin was progressively discontinued from the treatment during the following 24 days.”

Line 173, page 6: I found the phrase “the general stante was progressively degrading” confusing.  I suggest modifying to “his clinical condition progressively deteriorated”.

We modified accordingly.

DISCUSSION: This section is in need of some English language editing. There are some sections where the reader has difficulty undertanding

The discussion repeats phrases written in the introduction. I recommend reducing redundant phrases.

We improved our discussion section.

Line 185, Page 6: In relation to the ecological niche of the bacteria "everywhere" is not the correct scientific term. "Ubiquitous" is the correct term.

We revised it.

Line 254, Page 8: The sentence begins with a lowercase n, please correct this sentence.

We revised it.

Kind regards,

The authors

Reviewer 2 Report

I would like to thank the authors for the opportunity to review this case report of an extended brain abscess due to serratia in a neonate.

In this case report, a recurring medical problem among neonates is addressed. However, in reviewing the manuscript, some questions remained unanswered. Below the authors will find my comments and questions.

Figure 1 does not provide information that requires graphic interpretation. Reconsider this additional graphic.

Please explain the persistent need for ventilation over 16 days, was it caused by the anticonvulsant medication, in the sense of a sedative effect?

Table 1 was given a quantity definition in terms of bacterial growth. According to which definition was the antibiogram displayed? EUCAST?

Please reference Figure 4 in the text and explain the context of the figure, especially in the chronological relation of the course of treatment.

Figure 5 please expand the axis labeling in an adequate way to represent the scaling of both laboratory values and the time interval (days?).

With regard to Figure 6, please explain to what extent a daily spinal puncture was performed to obtain CSF and determine the laboratory values. In addition, adequate axis labeling would be beneficial.

Line 154 it would be exciting to know what led to the change to the mentioned antibiotic combination of cefotaxime, piperacillin-tazobactam, amikacin. Was this based solely on calculation?

If the external ventricular drainage was removed, (line 159) in what mode did the intraventricular antibiotic therapy occur (line 170f)?

Would it be possible to state the dosage of catecholamines (for example, substances, µg/kg/min)? The sole statement of maximum dosage is elusive and will vary between different centers.

Please shorten to historical initiation which is redundant to the introduction section.

Is there a double space hiding in line 187?

If the autors refer to the existence of references in the literature (line 192), they should also cite an appropriate reference.

The autors come to the statement: "Our opinion is that this is the mechanism in this case." regarding vaginal examinations. Is it known how often a vaginal examination was performed in the present case due to the complicated pregnancy? This would be a valuable clue to assess their assumption.

What do the authors understand by paraclinical tests? (line 234/235)

Regarding the comment on the prolonged administration of carbapenems (line 244), it would be interesting to know what the mode of application was used in the present case?

"I "Missing in line 255?

How do the authors conclude that aggressive antibiotic therapy caused the sepsis with candida?

Was the abscess punctured after initial imaging? And to the extent that it was not, what were the motivations?

The end of the discussion seems very abrupt.

Author Response

Dear reviewer,

We appreciate your comments. We have performed the modification according to your suggestions.

Figure 1 does not provide information that requires graphic interpretation. Reconsider this additional graphic.

At your suggestion, we decided to remove figure 1.

Please explain the persistent need for ventilation over 16 days, was it caused by the anticonvulsant medication, in the sense of a sedative effect?

Spontaneous breathing was impossible in severe neurological disease, and gradual weaning from mechanical ventilation was achieved.

Table 1 was given a quantity definition in terms of bacterial growth. According to which definition was the antibiogram displayed? EUCAST?

The diffusimmetric antibiogram was performed according to the EUCAST guide from 2010 using a Vitek2 Compact device (BioMerieux).

Please reference Figure 4 in the text and explain the context of the figure, especially in the chronological relation of the course of treatment.

A cystic cavity was formed in place of the abscess that communicated with the right lateral ventricle (Figure 4). Figure 4 represents the brain ultrasound performed on the 30th day of life.

Figure 5 please expand the axis labeling in an adequate way to represent the scaling of both laboratory values and the time interval (days?).

We modified.

With regard to Figure 6, please explain to what extent a daily spinal puncture was performed to obtain CSF and determine the laboratory values. In addition, adequate axis labeling would be beneficial.

CSF was obtained by lumbar puncture (the first two determinations), later, ventricular puncture was performed.

Line 154 it would be exciting to know what led to the change to the mentioned antibiotic combination of cefotaxime, piperacillin-tazobactam, amikacin. Was this based solely on calculation?

Antibiotics were changed according to modifications in CSF proteins and CSF glucose, blood values of CRP, and platelet count. Several classes of antibiotics have been tested. The timing and duration of administration are presented in Figures 5,6.

If the external ventricular drainage was removed, (line 159) in what mode did the intraventricular antibiotic therapy occur (line 170f)?

The external ventricular drain was replaced because we suspected a possible infection (Candida).

Would it be possible to state the dosage of catecholamines (for example, substances, µg/kg/min)? The sole statement of maximum dosage is elusive and will vary between different centers.

We have added the dosages.

Please shorten to historical initiation which is redundant to the introduction section.

We have improved our introduction section.

Is there a double space hiding in line 187?

We modified the error.

If the autors refer to the existence of references in the literature (line 192), they should also cite an appropriate reference.

We have added the appropriate references to our statements,

The autors come to the statement: "Our opinion is that this is the mechanism in this case." regarding vaginal examinations. Is it known how often a vaginal examination was performed in the present case due to the complicated pregnancy? This would be a valuable clue to assess their assumption.

Unfortunately, we are not aware of the exact number of vaginal examinations ( the obstetrician couldn’t provide this information) , but it was a complicated pregnancy, with prolonged hospitalization almost the entire pregnancy. We concluded, "A brain abscess identified in the first week of life is definitely due to an intrauterine infection. It is impossible to specify with certainty the moment of acquiring the intrauterine infection and the moment of the abscess formation. It could have been during the invasive maneuvers on the pregnant woman during the prolonged hospitalization, but infection during the assisted human reproduction procedure cannot be excluded”.

What do the authors understand by paraclinical tests? (line 234/235)

Through the paraclinical test , we referred to blood tests and CSF analysis. We have modified the text.

Regarding the comment on the prolonged administration of carbapenems (line 244), it would be interesting to know what the mode of application was used in the present case?

We used Meropenem 40mg/kg/dose every 12 hours in the first seven days of life, then every 8 hours, in one-hour infusion.

"I "Missing in line 255?

We modified it accordingly. 

How do the authors conclude that aggressive antibiotic therapy caused the sepsis with candida?

We have modified the manuscript. Besides the aggressive antibiotic therapy there were other factors involved.  “Neonatal risk factors for invasive Candida parapsilosis infections are low birth weight, prematurity, parenteral nutrition, the presence of central venous lines, catheters or other tubes, major surgery, and the use of broad-spectrum antibiotics, steroids or H 2 blockers”

Was the abscess punctured after initial imaging? And to the extent that it was not, what were the motivations?

We have added figure 1. Figure 1 presents the first brain ultrasound image from the second day of life, before the punction of the abscess. The abscess puncture was postponed, as the medical staff initially considered treating the abscess with medication.

The end of the discussion seems very abrupt.

We rephrased the discussion section.

Kind regards,

The authors

Round 2

Reviewer 2 Report

I would like to thank the authors for their extensive improvements and additions.